

# Comparative study of the efficacy of intra-arterial and intravenous transplantation of human induced pluripotent stem cells-derived neural progenitor cells in experimental stroke

Elvira Cherkashova[1,2], Daria Namestnikova[1,2], Georgiy Leonov[3], Ilya Gubskiy[1,2], Kirill Sukhinich[3], Pavel Melnikov[4], Vladimir Chekhonin[1,4], Konstantin Yarygin[3,5], Dmitry Goldshtein[6] and Diana Salikhova[7]

[1] Pirogov Russian National Research Medical University of the Ministry of Healthcare of Russian Federation, Moscow, Russian Federation

[2] Federal Center of Brain Research and Neurotechnologies of the Federal Medical Biological Agency of Russian Federation, Moscow, Russian Federation

[3] Orekhovich Research Institute of Biomedical Chemistry of the Russian Academy of Sciences, Moscow, Russian Federation

[4] Serbsky Federal Medical Research Centre of Psychiatry and Narcology of the Ministry of Healthcare of Russian Federation, Moscow, Russian Federation

[5] Russian Medical Academy of Continuous Professional Education of the Ministry of Healthcare of the Russian Federation, Moscow, Russian Federation

[6] Research Centre for Medical Genetics, Moscow, Russian Federation

[7] Institute of Molecular and Cellular Medicine, Medical Institute, RUDN University, Moscow, Russian Federation

Corresponding author
Diana Salikhova,
diana_salikhova@bk.ru

## ABSTRACT

**Background.** Cell therapy using neural progenitor cells (NPCs) is a promising approach for ischemic stroke treatment according to the results of multiple preclinical studies in animal stroke models. In the vast majority of conducted animal studies, the therapeutic efficacy of NPCs was estimated after intracerebral transplantation, while the information of the effectiveness of systemic administration is limited. Nowadays, several clinical trials aimed to estimate the safety and efficacy of NPCs transplantation in stroke patients were also conducted. In these studies, NPCs were transplanted intracerebrally in the subacute/chronic phase of stroke. The results of clinical trials confirmed the safety of the approach, however, the degree of functional improvement (the primary efficacy endpoint) was not sufficient in the majority of the studies. Therefore, more studies are needed in order to investigate the optimal transplantation parameters, especially the timing of cell transplantation after the stroke onset. This study aimed to evaluate the therapeutic effects of intra-arterial (IA) and intravenous (IV) administration of NPCs derived from induced pluripotent stem cells (iNPCs) in the acute phase of experimental stroke in rats. Induced pluripotent stem cells were chosen as the source of NPCs as this technology is perspective, has no ethical concerns and provides the access to personalized medicine.

**Methods.** Human iNPCs were transplanted IA or IV into male Wistar rats 24 h after the middle cerebral artery occlusion stroke modeling. Therapeutic efficacy was monitored

for 14 days and evaluated in comparison with the cell transplantation-free control group. Additionally, cell distribution in the brain was assessed.

**Results**. The obtained results show that both routes of systemic transplantation (IV and IA) significantly reduced the mortality and improved the neurological deficit of experimental animals compared to the control group. At the same time, according to the MRI data, only IA administration led to faster and prominent reduction of the stroke volume. After IA administration, iNPCs transiently trapped in the brain and were not detected on day 7 after the transplantation. In case of IV injection, transplanted cells were not visualized in the brain. The obtained data demonstrated that the systemic transplantation of human iNPCs in the acute phase of ischemic stroke can be a promising therapeutic strategy.

# INTRODUCTION

The importance of the search for new therapeutic strategies of ischemic stroke treatment is beyond doubt (*Campbell et al., 2019*; *Kim et al., 2020*; *Cherkashova et al., 2023*). In animal models of stroke, multiple preclinical studies have demonstrated remarkable therapeutic efficacy of the transplantation of various types of stem cells along with enhanced recovery of the affected animals (*Namestnikova et al., 2020*; *Zhang et al., 2021*; *Gao et al., 2022*). Mesenchymal stem cells (MSCs), bone marrow-derived mononuclear cells (BMMCs) and NPCs are probably the most safe, effective and qualified for clinical applications among all the studied cell types (*Chrostek et al., 2019*; *Singh et al., 2020*; *Rascón-Ramírez et al., 2021*).

Currently, the optimal cell type has not been defined yet, but in the vast majority of clinical trials, MSCs and BMMCs were used due to their availability, immunomodulation properties and the possibility of obtaining from many sources without ethical concerns (*Andrzejewska, Lukomska & Janowski, 2019*; *Kawabori et al., 2020*; *Lehnerer et al., 2022*; *Fauzi et al., 2023*). The results of conducted clinical studies demonstrated safety and feasibility of MSCs and BMMCs transplantation and a trend toward the enhancement of functional recovery after stroke (*Fauzi et al., 2023*), though in a part of trials the primary efficacy endpoints were non-significant (*Clark et al., 2017*; *Savitz et al., 2019*; *Song et al., 2022*).

Transplantation of NPCs as monotherapy (*Baker, Kinder & West, 2019*) or in combination with the other cell types (*Namestnikova et al., 2020*) is another tested therapeutic approach. NPCs are multipotent cells that can give all specialized cells of the nervous system (of both glial and neuronal lineages), making them attractive candidates for use in stroke cell therapy (*Hamblin et al., 2021*). NPCs can be isolated from the central nervous system of developing embryos, fetal, neonatal and adult brain, or derived *in vitro* from embryonic stem cells and induced pluripotent stem cells, as well as from somatic cells by direct transdifferentiation (*Tang, Yu & Cheng, 2017*). Transplantation of NPCs of

different origin in preclinical studies resulted in better functional outcomes for animals with experimental stroke (*Kokaia & Darsalia, 2018*; *Zhang et al., 2022*). Despite that the exact mechanism of NPCs therapeutic action remains not fully understood, it was shown that transplanted cells are able to reduce inflammation and blood brain barrier damage, increase neurogenesis, angiogenesis, neuroplasticity and even act as a cell replacement (*Baker, Kinder & West, 2019*; *Hamblin et al., 2021*).

Based on the success of preclinical studies, several clinical trials involving NPCs administration were initiated and finalized, and one (PISCES III) is still ongoing (*Kalladka et al., 2016*; *Chrostek et al., 2019*; *He, Sussman & Steinberg, 2020*). The number of studies of the NPCs transplantation is much lower compared to MSC cell therapy trials (*Fauzi et al., 2023*). This may be partially explained by ethical and safety concerns associated with the sources of NPCs, since in most clinical studies the fetal NPCs, immortalized NPC line or even carcinoma-derived neurons (NT2N cells) were administrated (*Pollock et al., 2006*; *He, Sussman & Steinberg, 2020*; *Kondziolka et al., 2000*). Transplantation of NPCs obtained from non-human sources (xenogeneic fetal porcine cells) was also tested. However, the trial was terminated due to the significant adverse events (*Dinsmore et al., 2005*). In all conducted studies, NPCs were transplanted intracerebrally in the subacute or chronic phase of stroke (*Kawabori et al., 2020*; *He, Sussman & Steinberg, 2020*). Summarizing the results of these clinical trials, it can be concluded that intracerebral transplantation of human NPCs was generally safe and caused minimal adverse events in non-acute stroke patients. In all studies, the authors reported about the tendency towards the improvement of the neurological deficit of stroke patients, however, as in the case of MSCs and BMMCs transplantation, the primary efficacy endpoint (the pre-determined and sufficient level of functional outcome) was not achieved in the majority of trials (*He, Sussman & Steinberg, 2020*). The reason for the insufficient effectiveness of NPCs therapy in randomized control clinical trials is debatable and may be related to suboptimal protocol of cell transplantation (cell dose, delivery route, time window, frequency of administration and others) and to not fully understood mechanism of action. More studies are needed in order to investigate the optimal transplantation parameters and among them, the most crucial one could be the timing of cell transplantation after the stroke onset (*Permana et al., 2022*). Many preclinical studies have demonstrated that transplantation of stem/progenitor cells could cause more prominent therapeutic effect when transplanted in the acute period of stroke (24–72 h) (*Yang et al., 2011*; *Wang et al., 2014*; *Toyoshima et al., 2015*). Despite the fact that intracerebral NPCs administration have proven to be relatively safe and well tolerated by patients in the subacute/chronic period of stroke, stereotaxic surgery and general anesthesia in the acute stroke patients may be associated with the elevated risk of complications (*Janowski, Wagner & Boltze, 2015*; *Muir et al., 2020*; *Fauzi et al., 2023*). Thereby, in cases of cell transplantation in the acute stroke, systemic (IA and IV) administration seems to be the most appropriate. It is worth noting that the data on systemic administration of NPCs to stroke patients is limited.

Since the access to the embryonic and fetal tissues is limited or even unacceptable in some countries (*Matthews & Moralí, 2020*; *Fernandez-Muñoz et al., 2021*), alternative sources of NPCs were found. The method of induced pluripotent stem cells (iPSCs)

induction introduced by *Takahashi & Yamanaka (2006)* provided the opportunity to obtain NPCs without ethical concerns and opened an era of personalized medicine. According to the results of conducted preclinical studies, the use of iNPC-based cell therapy demonstrated enhanced functional recovery after cell transplantation in preclinical rodent and pig stroke models (*Baker, Kinder & West, 2019*). In the majority of studies, iNPCs were transplanted intracerebrally/intrathecally while data on the systemic administration of this cells are limited (*Song et al., 2022*). Although basic animal studies showed that transplantation of iNPCs can be safe and efficacious, currently there is no published clinical study testing iNPCs in stroke patients (*Fernandez-Muñoz et al., 2021*). The main reason of the restriction of clinical application of iPSC-derived cells is the awareness of potential risk of tumorigenicity (*Volarevic et al., 2018*). However, the results of the first clinical trial for other severe diseases (leukemia, lymphoma, ataxia-telangiectasia, and others) indicated that carefully controlled iPSC-derived biomedical products can be safe in clinical settings (*Garitaonandia et al., 2016*; *Fernandez-Muñoz et al., 2021*) and several clinical studies applying iPSC-derived cells are ongoing and the results are expected soon (*Volarevic et al., 2018*). Further studies are needed in order to investigate the effectiveness of iNPCs therapy for ischemic stroke, especially in the acute phase.

Therefore, further preclinical and clinical studies are needed in order to investigate the effectiveness of the systemic iNPCs transplantation in ischemic stroke, especially in the acute phase. The present work was aimed to study the therapeutic effects of the IA and IV administration of iNPCs in the acute phase of experimental ischemic stroke in rats.

## MATERIALS & METHODS

### Cell culture

iPSCs were derived according to the protocol described previously (*Salikhova et al., 2020*). Donor skin biopsy collection was approved by the Institutional Ethics Committee of the Research Centre for Medical Genetics (Protocol No. 2019-2/3 from October 13, 2020). All patients given written informed consent for using the tissue for research purposes. CTS CytoTune-iPS 2.1 Sendai reprogramming kit (Invitrogen, Carlsbad, CA, USA) were used for fibroblast reprogramming. Obtained iPSCs were cultivated in Essential E8 Medium (Gibco, Waltham, MA, USA) using Petri dishes covered by recombinant vitronectin (10 μg/mL, Gibco, Waltham, MA, USA). iPSCs were detached upon reaching a culture of 80% confluency by incubation in Versen solution (PanEco, Berg am Irchel, Switzerland) for 5 min at 37 °C. The cell suspension was centrifuged at 800 rpm for 5 min and the pellet was transferred to new Petri dishes with the addition of ROCK inhibitor (5 μM, Merck Millipore, Rahway, NJ, USA) to the culture medium for 24 h.

The pluripotent status of the iPSCs was confirmed, as previously described (*Salikhova et al., 2021*), by their ability to differentiate into cells belonging to all three germ layers, to form embryoid bodies, and to express the pluripotent markers Oct4, Nanog, SSEA4, TRA-1-81. iPSCs were transferred as colony fragments to 24-well Ultra Low Adhesion Plates (Corning, Corning, NY, USA) and cultured in DMEM/F12 medium supplemented 20% FBS, 2 mM glutamine, 1% mixture of essential amino acids, 100 mg/L penicillin-streptomycin

(Paneko, Moscow, Russia) for formation of embryoid bodies. The formed embryoid bodies were transferred to Petri dishes coated with 0.1% gelatin (Paneko, Moscow, Russia) for observation of cell migration after 2–3 weeks of cultivation. The obtained extensive areas of differentiated cells were fixed and immunocytochemical analysis was performed for markers of three germ layers using primary antibodies against pan-cytokeratin (ectoderm marker, ab7753, Abcam, Boston, MA, USA), vimentin (mesoderm marker, ab92547, Abcam, Boston, MA, USA), α-fetoprotein (endodermal marker, ab3980, Abcam, Boston, MA, USA). To confirm the phenotype of iPSCs, the immunocytochemical analysis on markers of pluripotency was performed using primary antibodies to NANOG, SSEA4, OCT4, TRA-1-81 (StemLight Pluripotency Antibody Kit, Cell Signaling Technology, San Antonio, TX, USA).

iNPCs were derived from iPSCs using the mix of small molecules: 10 μM SB431542 (Tocris, USA), 2 μM Dorsomorphine (Sigma Algrich, St. Louis, MO, USA), and 200 nM LDN193189 (Sigma Algrich, St. Louis, MO, USA) which were TGF-β and BMP signaling inhibitors. The differentiation lasted for 2 weeks on Petri dishes covered by Matrigel (Corning, Corning, NY, USA). The formed radially organized cellular structures (also called neural rosette) were subcultured with Versen solution for 5 min at 37 °C. The cell suspension was centrifuged at 1,200 rpm for 5 min. The obtained pellet of the iNPCs was cultivated in the DMEM/F12 medium (PanEco, Berg am Irchel, Switzerland) supplemented by 2% of the B27 (Gibco, Waltham, MA, USA), 2 mM glutamine (PanEco, Berg am Irchel, Switzerland), 100 mg/L penicillin-streptomycin (PanEco, Berg am Irchel, Switzerland), and 10 ng/ml FGF-2 (ProSpec, UK) using Matrigel substrate. The phenotype of the obtained iNPCs confirmed immunocytochemically using primary antibodies against Nestin (ab105389, Abcam), PAX6 (ab5790, Abcam, Boston, MA, USA), and Sox2 (ab79351, Abcam, Boston, MA, USA).

## Immunocytochemistry

iPSCs and their derivate cultures were fixed in 4% paraformaldehyde solution (PanEco, Berg am Irchel, Switzerland) for 10 min at room temperature; washed with PBS; pre-incubated in 0.25% Triton X-100 and 1% BSA in PBS for 30 min; and incubated with primary antibodies for 60 min in the dark. Secondary antibodies (anti-mouse IgG conjugated to Alexa Fluor 555 (A-21422, Invitrogen, Carlsbad, CA) or anti-rabbit IgG conjugated to Alexa Fluor 488 (A-11008, Invitrogen, Carlsbad, CA) was applied for 60 min in the dark. The nuclei was counterstained with DAPI (Sigma-Aldrich, , St. Louis, MO, USA) solution (1 μg/mL in PBS). The images were recorded with an Axio Observer.D1 inverted fluorescence microscope equipped with Axio-Cam HRc camera (Carl Zeiss, Jena, Germany).

## Cell labeling

The transplanted iNPCs were pre-labeled with the GFP protein using the LVT-TagGFP lentiviral vector (Evrogen, Moscow, Russia) for visualization. The virus suspension was added to the culture medium with 4 μg/ml polybrene (Sigma-Aldrich, St. Louis, MO, USA) according to the manufacturer's instructions. The cells were washed twice with the phosphate-buffered saline (PanEco, Moscow, Russia) after 12 h incubation and the

maintenance culture medium was added. The iNPCs were additionally labeled with the PKH26 Red Fluorescent Cell Linker kit (Sigma-Aldrich, USA) following the instructions of the manufacturer. Briefly, the cells were harvested using Versen solution (PanEco, Moscow, Russia), centrifuged at 1,200 rpm for 5 min, washed with Hank's Balanced Salt Solution (HBSS), and re-suspended in 1 ml of the Diluent C reagent. The cell suspension was mixed with an equal volume of the labelling solution (4 nM PKH26), and incubated at room temperature for 5 min. The staining process was stopped by the addition of 2 ml of fetal bovine serum. Cells were washed twice with HBSS, and re-suspended in the phosphate-buffered saline (PanEco, Moscow, Russia). The obtained suspension was analyzed 72 h post-transfection using a CyFlow ML flow cytometer (Partec, München Germany), and the number of labeled cells was counted using the FloMax software.

## Animals

Adult male Wistar rats weighing 250–300 g ($n = 69$) were purchased from "SMK STESAR" (Vladimir, Russian Federation). The experimental animals were housed 4 to 5 per cage under standard housing conditions (12-h/12-h light/dark cycle, room temperature $22 \pm 2$ °C, humidity 45–65%) with free access to standard rodent chow and water. Enrichment based on no risk to the animals (*i.e.,* cause injuries or excessive aggression), to the humans (*i.e.,* jeopardize the health and safety of the animal staff), or to the experiments (*i.e.,* cause undesirable interference or an excessive increase in the number of animals used). All animal experiments were carried out in accordance with the guidelines of the Declaration of Helsinki and the Directive 2010/63/EU on the protection of animals used for scientific purposes of the European Parliament and the Council of European Union dated 22 September 2010 and were approved by the Pirogov Russian National Research Medical University Animal Care and Use Commission (protocol code No 24/2021 from 10 December 2021). In vivo studies are reported according to the ARRIVE guidelines (v. 2.0). Surgery was performed under inhalation anesthesia with the mixture of 2,5–3% isoflurane (Aerrane, Baxter HealthCare Corporation, Deerfield, IL, USA) and of 97–98,5% atmospheric air using an animal anesthesia system (E-Z-7000 Classic System, E-Z-Anesthesia® Systems, Palmer, PA, USA). After the stroke modeling using endovascular middle cerebral artery occlusion method, the animals were kept separately in a cage with a 37 °C heating pad to avoid additional injury and for better recovery after surgery. The animals were intraperitoneally administered with 2 ml of saline solution in order to avoid hypovolemia in the first hours after surgery, when the mobility of the animal would be limited due to the severity of the condition. At the end of the observation period (14 days after cell transplantation) and for histological examination (2 h, 3 and 7 days after transplantation) the experimental animals were euthanized by inhalation anesthesia with a lethal dose of isoflurane and by injection of a lethal dose of Zoletil.

## Animal study design

The experimental ischemic stroke was induced in rats ($n = 69$) using the model of transient middle cerebral artery occlusion (MCAO). The animals were tested using the Modified Neurological Severity Scores (mNSS) 24 h after stroke modeling, before cell administration,

and MRI study was performed in order to assess the severity of the rats' condition, the volume of brain damage, and the exclusion of complications of the model. Rats with hemorrhagic complications were removed from the experiment in order to objectify the evaluation of the data. Then animals were randomly divided into three groups: control group—rats without cell therapy ($n = 21$); iNPCs IA—rats with intra-arterial transplantation of iNPCs ($n = 22$, 11 for evaluation of the therapeutic efficiency and 11 for the study of cell distribution); iNPCs IV—rats with intravenous transplantation of iNPCs ($n = 26$, 15 for evaluation of the therapeutic efficiency and 11 for the study of cell distribution). Only the person responsible for the administration knew the classification of the groups.

The therapeutic efficiency of iNPCs transplantation was estimated by the survival rate, neurological deficit and stroke volume of animals within a 14 days period. The cells' distribution was evaluated by histological examination of the rat brain. For histochemical studies, rats were sacrificed at 2 h, 3 and 7 days after iNPCs transplantation using the method described above. All manipulations (tests, cell administration and histological examination) were assessed for all groups at the same time of the day to minimize potential confounders, such as the order of treatments and measurements or animal/cage location and treatment groups.

## Stroke modeling

The experimental ischemic stroke was induced using endovascular transient (90 min) MCAO developed by *Koizumi (1986)* and modified by Longa (*Longa et al., 1989*) as was described in detail previously (*Gubskiy et al., 2018*). The accuracy of the MCAO and possible complications were monitored by MRI guidance. The endovascular occlusion of the middle cerebral artery was carried out by a blunt-ended silicon coating monofilament (Doccol Corporation, Sharon, MA, USA; diameter of filament was 0.19 mm, length 30 mm, diameter of the tip 0.37+0.02 mm, length of the tip 3–4 mm). After removing the filament and suturing the surgical wound the animals were injected 2 ml of saline intraperitoneally and 0.2 ml of gentamicin intramuscularly and placed in a heated cage to recover from anesthesia.

## Cell transplantation

IA and IV cell transplantation was performed 24 h after stroke modeling as previously described (*Gubskiy et al., 2022*). For the IA administration, the bifurcation of the right common carotid artery was isolated and the pterygopalatine artery was ligated. A microcatheter (rodent tail vein catheter with 1F (1/3 mm) diameter and 28 cm length, Braintree Scientific, Inc., Braintree, MA, USA) filled with saline was inserted through the stump of the external carotid artery into the lumen of the common carotid artery for 5–6 mm in the direction opposite to the blood flow. The transplantation of iNPCs was performed through the microcatheter with the use of microinjector (Leica Microsystems GmbH, Braintree, MA, Germany) at the speed of 100 µl/min. Importantly, the blood flow in the common and internal carotid arteries was maintained during the transplantation to avoid cerebral embolism. For the IV cell transplantation, the right femoral vein was exposed

and a microcatheter (rodent tail vein catheter with 1F (1/3 mm) diameter and 28 cm length, Braintree Scientific, Inc., Braintree, MA, USA) was inserted. The transplantation of iNPCs was performed at the speed of 250 µl/min to avoid pulmonary embolism. The density of cell suspensions for IA and IV transplantations was $7 \times 10^5$ and $2 \times 10^6$ of iNPCs in 1 ml of phosphate-buffered saline respectively. For the rats from the groups for the evaluation of the therapeutic efficiency unlabeled iNPCs were administrated in order to avoid the potential influence of the labels to the cells' effect. Labeled cells were transplanted only into the rats for the histological study for evaluation of cell distribution.

## Neurological outcomes

The neurological deficit was estimated using the Modified Neurological Severity Scores (mNSS) for animals with stroke on the 1st, 7th and 14th day after transplantation. The mNSS combines a set of tests and allows to evaluate motor and sensory functions, reflexes and balance (*Schaar, Brenneman & Savitz, 2010*). In each test one point is given for failure and no points for success. The severity of stroke is estimated by the sum of points scored. According to the mNSS scale the maximum number of points is 18, which corresponds to the most severe neurological deficit, 12–18 points estimated as severe stroke, 8–12–moderate stroke, 1–7–mild stroke.

## MRI

The magnetic resonance imaging (MRI) was carried out using a 7T ClinScan system for small animals (Bruker BioSpin, USA) under isoflurane inhalation anesthesia (3.5–4% isoflurane mixed with pure oxygen). To perform stroke volume morphometry, T2-weighted imaging was carried out at the 1st, 7th and 14th day after transplantation. MRI morphometry was performed using ImageJ software (Rasband, W.S., ImageJ, National Institutes of Health, Bethesda, Maryland, USA). The volume of the stroke (V) was analyzed based on T2-weighted images by the summation of volumes measured in adjacent cross sections according to the following formula: $V = (S1 + \ldots + Sn) \times (h + d)$, where $S1, \ldots, Sn$ is area measured on slice, h is the slice thickness, and d is the gap interval between slices.

## Immunohistochemistry

Animals were sacrificed at 2 h, 3 and 7 days after the IA or IV transplantation of iNPCs by inhalation anesthesia with a lethal dose of isoflurane and additional injection of a lethal dose of Zoletil. After transcardial perfusion using 4% paraformaldehyde (PFA), the brains were removed and post-fixed at 4 °C overnight in the same fixative, washed with PBS and cryoprotected in the 30% sucrose solution. The coronal sections (40 µm thick) were obtained at a cryostat microtome (Leica CM1900, Munich, Germany). The sections were collected and stored for subsequent immunohistochemistry. The sections were incubated in a mixture of 5% normal goat serum (Sigma-Aldrich, St. Louis, MO, USA), 0.3% Triton x-100 (Sigma-Aldrich, St. Louis, MO, USA) and primary antibodies anti-RECA (rat endothelial cell of the blood–brain barrier antibody) (1:200, ab264524, Abcam, Cambridge, UK) or anti-EBA (endothelial cells of the blood–brain barrier antibody) (1:100, Biolegend, San Diego, USA) in 0.01 M PBS (pH 7.4) at 4 °C overnight. Then sections were rinsed with PBS and incubated with secondary antibodies (1:500, anti-mouse IgG Alexa fluor

647, Invitrogen, USA) for 2 h at room temperature. Nuclei were counterstained with DAPI solution (2 µg/mL, Sigma-Aldrich, St. Louis, MO, USA). Fluorescence confocal micrographs were taken with the Nikon A1R MP laser scanning confocal microscope (Nikon Instruments Inc., Tokyo, Japan). For the group of rats with IA iNPCs administration the morphometric analysis of obtained images was carried out. The percentage of double-labeled cells was calculated on the assumption of the total number of cells (cell nuclei) in the histological section. The percentage of GFP+ and PKH26+ cells was calculated according to the following formula: (labeled cells/total number of cells) × 100%.

## Statistical analysis

Statistical analysis and data visualization were performed using Python 3.10 (*Van Rossum & Drake, 2009*) in Jupyter Notebook (*Kluyver et al., 2016*). The normality of distribution was evaluated by Shapiro–Wilk's test. Animal survival was assessed by the Kaplan–Meier method using the Log Rank test (*Davidson-Pilon, 2019*). Two-way ANOVA was used to evaluate the dynamics changes of the neurological deficit and stroke volume of experimental animals. The values of mNSS score and stroke volume obtained on day 7 and 14 were normalized based on the data obtained on day 1. The pairwise comparison of the groups with the FDR correction with Benjamini–Hochberg method was performed. The final results are presented after the pairwise comparison. The significance level was set at 0.05.

The sample size for the animal experiments was estimated based on the results of our previous study of the therapeutic effects (mNSS) of neural precursor cells after IA transplantation in MCAO rats (*Namestnikova et al., 2021*). The sample size was initially estimated on the assumption of power $= 0.8$ and significance level $p = 0.05$ for three groups with three measurements, and the calculations showed that the number of rats in each group should be at least $n = 6$. Originally, the number of rats in each group was planned to be $n = 12$ (two times more than calculated sample size), however, considering the high mortality of rats from the control group (rats with experimental stroke without any treatment) and the necessity to withdrawal rats for histological examination from the groups with iNPCs transplantation, the number of rats in each group was increased and was 21, 22 and 26 for the control group, IA and IV group respectively.

## RESULTS

### Cell culture characterization

Human iPSCs formed smooth edged colonies morphologically similar to embryonic stem cells (ESCs). The colonies consisted of tightly packed cells with high nucleus-cytoplasm ratio, immunopositive for the following pluripotency markers: TRA-1-81, SSEA4, Oct4, Nanog (Fig. 1A). The cells had the ability to form embryoid bodies, and spontaneously differentiate into the derivatives of three germ layers –mesoderm (vimentin+ cells), ectoderm (pan-cytokeratin+ cells), and endoderm ($\alpha$-fetoprotein+ cells) (Fig. 1B). Neural differentiation of iPSCs was induced by dual SMAD inhibition that led to formation of neural rosettes. The obtained iNPCs were immunopositive for neural markers Pax6, Nestin, and Sox2 (Fig. 1C). For the *in vivo* cell tracking, iNPCs were transduced by lentiviral vector expressing the GFP and additionally stained by the lipophilic dye PKH26.

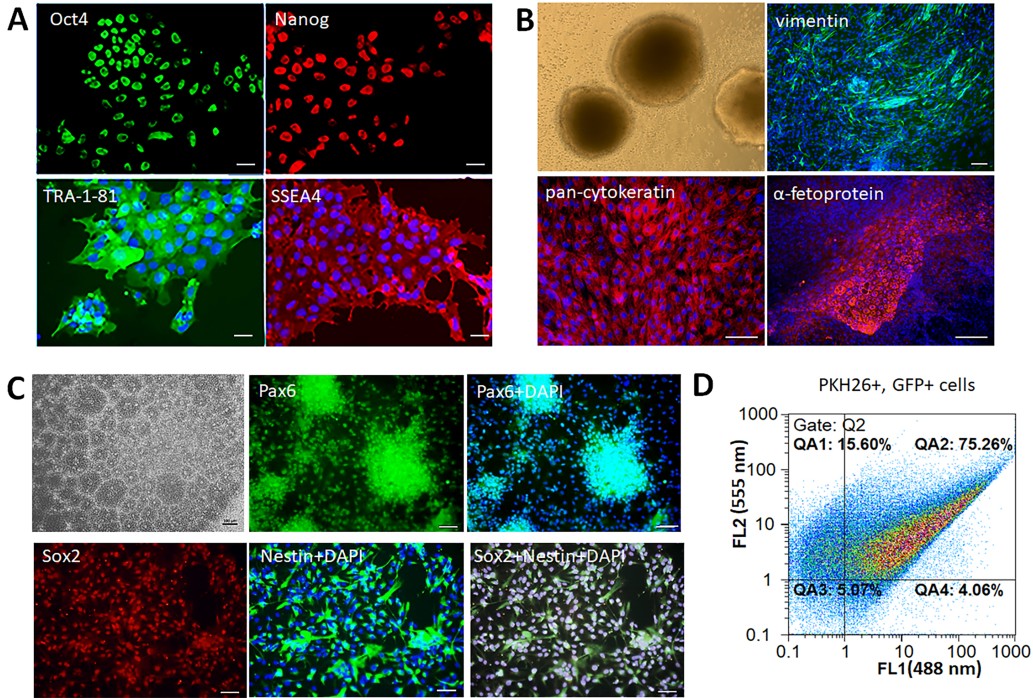

**Figure 1 Characterization of the iPSCs and iNPCs cultures.** (A) Immunocytochemical analysis of the iPSCs for the pluripotency markers Oct4, Nanog, TRA-1-81, and SSEA4. The nuclei were counterstained with DAPI. (B) iPSCs derived embryoid bodies (phase-contrast microscopy) and their differentiation into the three germ layers derivatives –mesoderm (vimentine$^+$ cells), ectoderm (pan-cytokeratin$^+$ cells), and endoderm ($\alpha$-fetoprotein$^+$ cells). (C) Characterization of iNPCs: phase-contrast microscopy, immunocytochemical analysis for neuronal markers Pax6, Nestin, and Sox2. Scale bar, 100 μm. (D) Flow cytometry analysis of GFP$^+$ and PKH26$^+$ cells, the efficiency of double cell labeling was 75.3 ± 5%.

The efficiency of transduction and cell labeling were determined by flow cytometry. The fluorescence intensity of transduced (GFP+) cells presented on the abscissa (FL1 channel), and PKH26+ labeled cells on the ordinate (FL2 channel). Accordingly, the efficiency of iNPCs transduction was 79.3 ± 4% (gates QA2 and QA4), and efficiency of PKH26+ labeling cells was 91 ± 6% (gates QA1 and QA2). The percentage of double-labeled cells was 75.3 ± 5% (gate QA2) (Fig. 1D).

## Therapeutic efficiency of the iNPCs transplantation

The survival of animals after iNPCs transplantation was estimated over the period of 14 days. The Kaplan–Meier survival curves are presented at Fig. 2A. According to the results of statistical analysis, the survival rate of animals from therapy groups was significantly higher compared to the control group. No significant differences were found between groups with different routes of iNPCs transplantation. Both IA and IV administration of iNPCs improved animal survival rate ($p \leq 0.05$) compared to the control group.

The neurological outcomes were evaluated using the mNSS on the 1st, 7th and 14th day post-transplantation. As seen from Fig. 2B, the dynamics of changes of the neurological deficit in rats getting the IA or IV iNPCs transplantation differed significantly from that

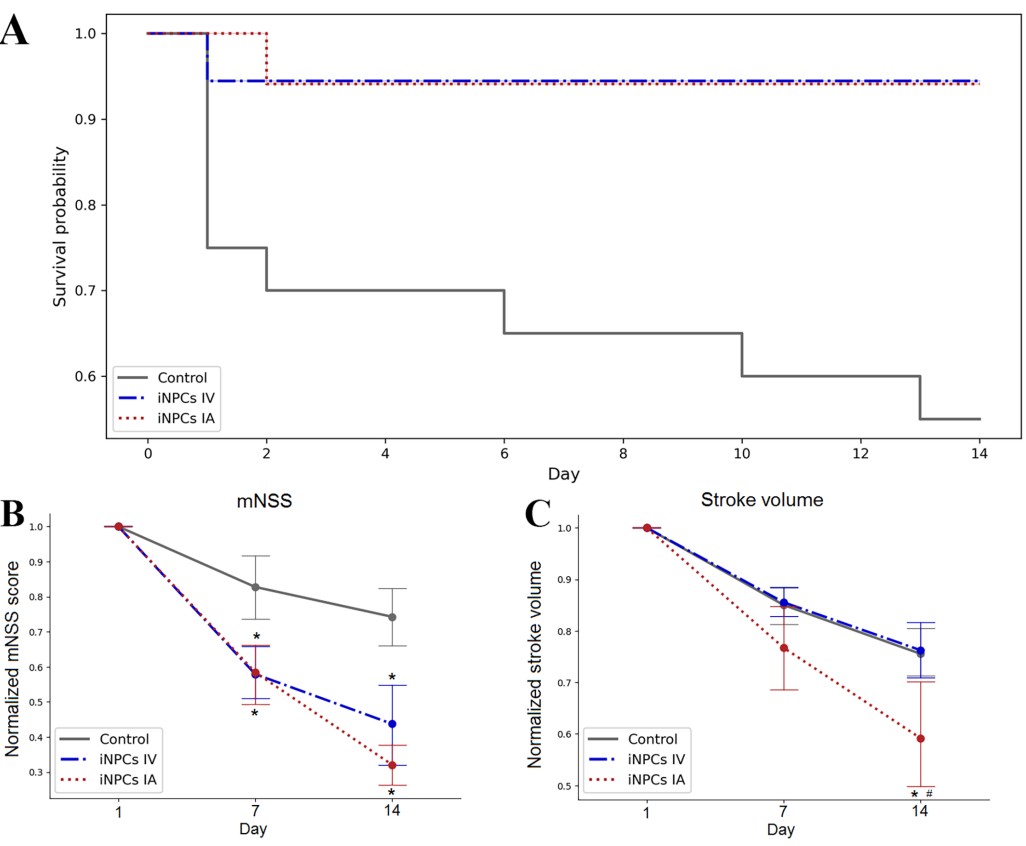

**Figure 2 The therapeutic efficiency of iNPCs transplantation.** (A) Kaplan–Meier survival curves for experimental groups during a 14 days observation period. Both IV and IA iNPCs administration significantly improves the survival of experimental animals. (B) The dynamics of changes of the neurological deficit according to mNSS in the experimental groups. The mNSS score was estimated on the 1st, 7th and 14th day after transplantation. The groups with IA and IV iNPCs administration did not differ significantly between each other, but differed significantly from the control group. (C) The dynamics of changes of the stroke volume estimated by the T2WI MRI during the 14 days period. The group with IA iNPCs transplantation differed significantly from the iNPCs IV and control groups. The data are presented as mean ± SD. Asterisks (*) indicate significant differences ($p \leq 0.05$). Hashes (#) indicate significant differences ($p \leq 0.05$) between intra-arterial and intravenous transplantation.

in the control group ($p \leq 0.05$). However, no significant difference was revealed between groups with different ways of iNPCs transplantation. Therefore, both routes of the systemic administration of iNPCs seem equally effective.

The stroke volume was evaluated by MRI on the 1st, 7th and 14th day after transplantation of iNPCs or vehicle (Fig. 2C). Only IA iNPCs transplantation led to significantly faster and pronounced reduction of the stroke volume compared to the control group during the observation period ($p \leq 0.05$). No significant difference in the dynamics of stroke volume changes was found between the group with IV iNPCs administration and the control group.

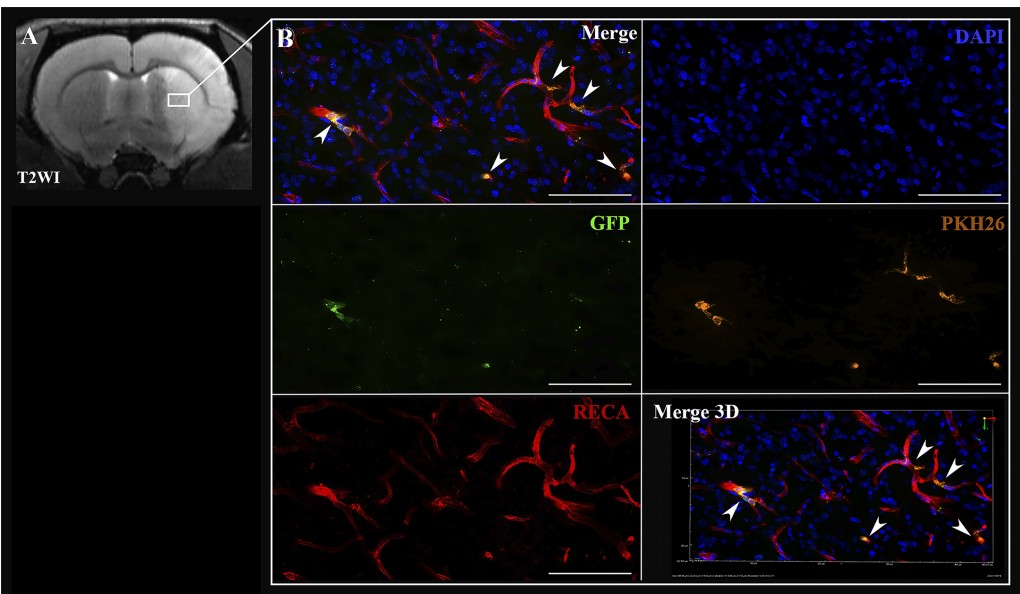

**Figure 3** **Visualization of iNPCs in the brain 2 h after the IA transplantation.** (A) The T2-weighted brain MRI demonstrated the presence of the ischemic lesion (hyperintense zone in the right hemisphere). (B) Confocal fluorescence images of rat brain from the area marked on (A). iNPCs double labeled with PKH26$^+$ (orange) and GFP$^+$ (green) cells visualized in the cerebral blood vessels (immunohistochemically stained in red colour by the anti-rat endothelial cell antibody (RECA)). Nuclei contrasted with DAPI (blue). Scale bar, 50 μm. Bottom right insert: 3D-reconstruction of z-stacks demonstrated the iNPCs inside cerebral vessels. Double labeled cells are marked with the white arrow heads on merge images.

## Cell distribution

In case of the IA administration of iNPCs, labeled cells were detected inside cerebral blood vessels 2 h after transplantation (Fig. 3). Transplanted cells were distributed in the right hemisphere in the infarct area and the peri-infarct zone.

Transplanted cells were found 2 h after IA administration in all experimental animals ($n = 3$). The percentage of such cells was $2.2 \pm 0.3\%$ per histological section of the rat brain. Subsequently, 72 h after IA administration just single cells remained inside the cerebral blood vessels and were detected only in one rat from the experimental group (the total number $n = 4$). Therefore, it can be concluded that the number of transplanted labeled cells decreased over time (the example is given in Fig. 4). As can be seen from Fig. 4A 2 h after IA administration the transplanted iNPCs were detected in several blood vessels, whereas after 72 h (Fig. 4B) only single labelled cells were observed. Remarkably, within 72 h after cell administration human iNPCs were visualized inside cerebral blood vessels in contact with their inner wall (vessel were immunostained for the endothelial barrier antigen (EBA)) and the migration of transplanted cells through the structure of the blood–brain barrier into the brain parenchyma were not observed. After 7 days post-transplantation labeled iNPCs were not detected in the brain in all experimental animals ($n = 4$).

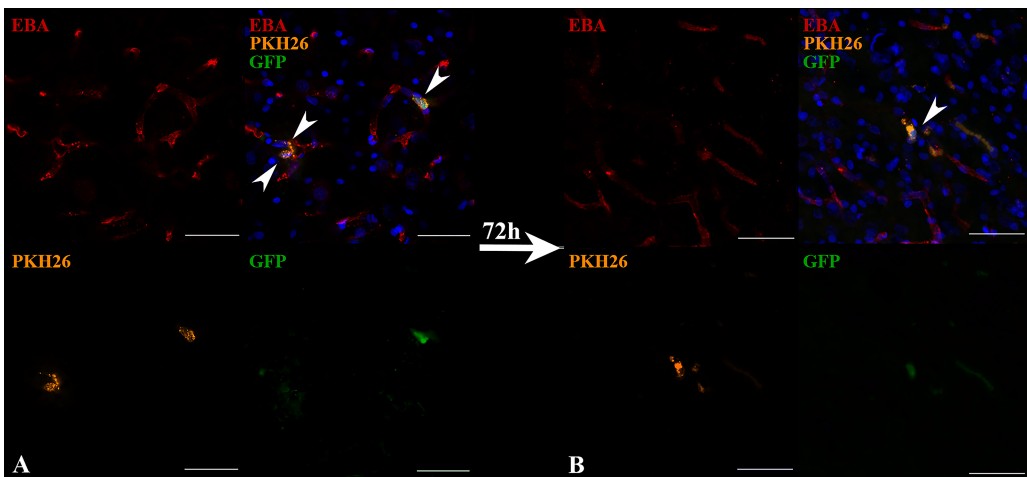

**Figure 4 The reduction of the number of transplanted iNPCs in the rat brain 72 h after IA administration.** Confocal fluorescence images of rat brain 2 h (A) and 72 h (B) after the IA transplantation are presented. iNPCs double labeled with PKH26+ (orange) and GFP+ (green) were detected inside cerebral blood vessels (immunohistochemically positive for the endothelial brain antigen (EBA)). Nuclei contrasted with DAPI (blue). Scale bar, 50 µm.

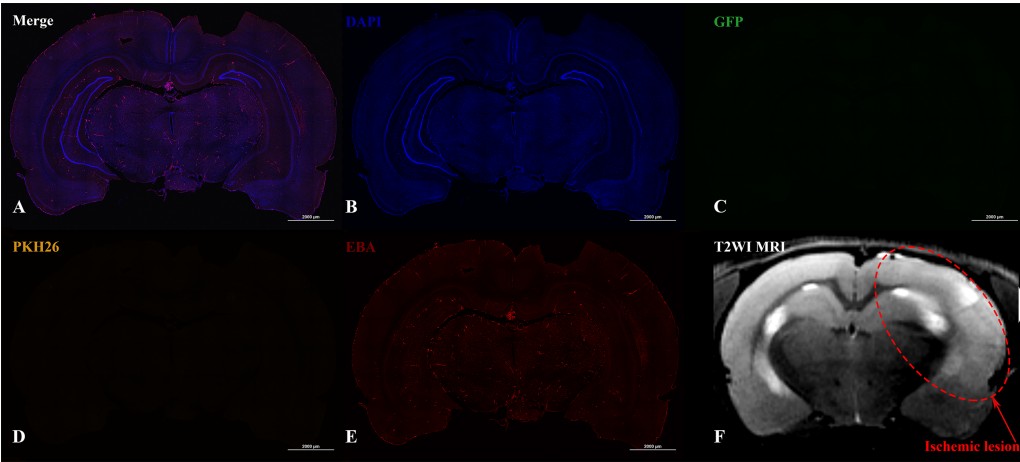

**Figure 5 Histological images and MRI of the rat brain with experimental ischemic stroke 2 h after iNPCs IV transplantation.** (A–E) Confocal fluorescence images of the brain slices. GFP+ (C) and PKH26+ (D) cells were not detected in the brain parenchyma and inside cerebral vessels stained with endothelial barrier antigen, EBA (E). Scale bar, 2,000 µm. (F) - T2-weighted MRI, the red dotted line indicates the ischemic lesion.

In the case of IV administration of iNPCs, labeled cells (GFP+ and PKH26+) were not detected in the brain 2 h ($n = 4$) after injection, as well as on the 3rd ($n = 4$) and 7th day ($n = 3$) after transplantation (Fig. 5).

## DISCUSSION

In this study, systemic (IA and IV) transplantation of iNPCs into rats 24 h after experimental ischemic stroke modeling were performed. Both IA and IV administration of iNPCs in the acute phase of ischemic stroke reduced the mortality of experimental animals and contributed to the improvement of neurological deficit. Therefore, the obtained results have confirmed the effectiveness of iNPCs systemic administration. Although there was no significant difference between the groups with IA and IV cell administration in the dynamics of the neurological deficit reduction, at the day 14 post-transplantation there was a tendency towards a more pronounced effect after IA cell transplantation. The revealed trend makes it relevant to conduct further studies with larger samples and a large number of tests to clarify this issue. At the same time, according to the MRI data the IA administration was more efficient and led to faster and more prominent reduction of the stroke volume.

In the vast majority of studies, the therapeutic efficacy of NPCs was estimated after intracerebral transplantation, while the information of the effectiveness of systemic administration is limited (*Kokaia & Darsalia, 2018*; *Baker, Kinder & West, 2019*; *Surugiu et al., 2019*; *Zhang et al., 2022*; *Yang et al., 2022*). The intracerebral stereotaxic administration of NPCs in the acute phase of stroke may be challenging, while systemic administration seems more appropriate. This assumption is supported by the results of *Doeppner et al. (2015)*, who tested six different routes of NPCs delivery and concluded that systemic, especially IV transplantation, is an attractive and effective strategy for stroke therapy. IV transplantation of iNPCs was used in several other studies. In the study of *Watanabe et al. (2016)*, human iNPCs were intravenously transplanted into rats 6 h after MCAO. The authors observed better functional recovery of animals and neuroprotective effects, which they attributed to better regulation of early inflammatory events in the cerebral ischemia condition. *Doeppner et al. (2014)* also demonstrated the improvement of post-stroke functional recovery after IV transplantation of murine adult subventricular zone-derived NPCs in the acute periods of stroke. The authors explained the positive therapeutic effects by the stabilization of the blood brain barrier and the modulation of immune response. In this study, the impact of IV iNPCs transplantation on the rate of the reduction of stroke volume was not revealed. At the same time *Cheng et al. (2015)* showed better neurological outcome without the reduction of cerebral infarct volume after IV administration of a mouse-derived cell line of NPCs after transient ischemic stroke in adult rats, which is in accordance with the results of the current study. The reason for different effects of NPCs transplantation on the stroke volume reported by various authors remains unclear.

In our study, IV transplantation of iNPCs significantly improved the survival rate and neurological deficit of the experimental animals compared to the control group. Despite the significant therapeutic effects after IV iNPCs administration, transplanted cells were not detected in the brain within the observation period (14 days). The absent or poor cell engraftment in the brain has been demonstrated for different types of stem/progenitor cells after IV administration and may be partially explained by the entrapment of transplanted cells in peripheral organs (*Lappalainen et al., 2008*; *Fischer et al., 2009*; *Sanchez-Diaz et al., 2021*; *Nose et al., 2021*; *Cherkashova et al., 2023*). Interestingly, there are evidences that

transplanted cells may mediate their therapeutic effects targeting internal organs, for instance lungs (*De Witte et al., 2018*) or spleen (*Acosta et al., 2015*; *Wang et al., 2019*; *Xu et al., 2019*). Moreover, NPCs are capable of releasing of many different soluble factors into the bloodstream (paracrine effect), which can mediate post-stroke recovery (*Willis et al., 2020*). There are many potential mechanisms of paracrine cells' action and among them the neuroprotection, angiogenesis, and immunomodulation are the most feasible (*Andres et al., 2008*). Notably, all these mechanisms are based on the assumption that transplanted cells through releasing of different factors and molecules (for instance vascular endothelial growth factor (VEGF), brain-derived neurotrophic factor (BDNF), nerve growth factor (NGF), epidermal growth factor (EGF), insulin-like growth factor (IGF)) act on the surviving of neurons of the host brain tissue, as well as on the glial and immune cells. Some experiments have shown, that IV administration of NPCs reduced inflammation and tissue injury without reaching the brain, and provided a peripheral immunosuppression by bystander inhibitory effect on T cell activation and proliferation in lymph nodes (*Ben-Hur, 2008*; *Hermann et al., 2014*; *Kokaia, Llorente & Carmichael, 2018*). In another study the authors discovered the ability of NPCs to interact with the organs of the immune system that led to the reduction of neurologic deficit and brain edema after IV NPCs administration in a rat model of hemorrhagic stroke (*Lee et al., 2008*). They screened inflammatory cytokine levels in spleen and lymph nodes, and found that NPCs attenuated the splenic inflammatory activations including TNF-α, IL-6, and NF-κB. NPCs also decreased the number of TNF-α-expressing macrophages in spleen. Thus, authors suggested, that IV NPCs administration can attenuate systemic inflammatory response after stroke and protect the brain by indirect mechanism (*Lee et al., 2008*). Some studies demonstrated NPCs - induced neuroprotection and angiogenesis. The neuroprotective effects accompanied by increased *in vivo* bioavailability of neurotrophins such as nerve growth factor (NGF), brain-derived neurotrophic factor (BDNF), ciliary neurotrophic factor (CNTF), and glial-derived neurotrophic factor (GDNF) by IV transplanted NPCs. Induced the proliferation of existing vascular endothelial cells (angiogenesis) and mobilization and homing of endogenous endothelial progenitors (vasculogenesis) by increasing the levels of angiogenic factors, such as vascular endothelial growth factor (VEGF), fibroblast growth factor-2 (FGF-2), and chemoattractant stromal cell-derived factor - 1 (SDF-1) (*Andres et al., 2008*; *Hamblin et al., 2021*).

Targeted cell delivery to the brain circulation can be achieved by IA transplantation (*Malysz-Cymborska et al., 2021*; *Guzman, Janowski & Walczak, 2018*; *Namestnikova et al., 2021*). Nowadays, this route of administration may be successfully translated to the clinical settings due to the widespread use of endovascular treatment of acute ischemic stroke (*Griauzde et al., 2019*; *Powers et al., 2019*; *Berge et al., 2021*). It should be noted that the IA cell infusion may lead to cerebral embolism and requires precise selection of the infusion parameters (*Boltze et al., 2015*; *Guzman, Janowski & Walczak, 2018*). The most important determinants of the safety of stem cell transplantation are the cell size and speed of cell infusion (*Janowski et al., 2013*). For example, it was shown that the IA transplantation of MSCs due to their large size (up to 25 μm) requires very precise selection of infusion parameters in order to prevent overdosing and cerebral cell embolism, while IA infusion

of glial progenitor cells is considered to be more safe as the cell size is relativity small (up to 10 μm) (*Janowski et al., 2013*; *Porterfield, 2020*). The iNPCs assessed in this study have the same cell diameter as the glial progenitor cells. Nevertheless, the particular attention was paid on the safety of cell transplantation and therefore the infusion parameters that were selected based on the literature data (*Guzman, Janowski & Walczak, 2018*) and the results of our previous works (*Namestnikova et al., 2017*; *Namestnikova et al., 2021*; *Gubskiy et al., 2022*). Moreover, in the current study, the safety of the IA administration was controlled by performing DWI MRI before and after cell infusion, and no embolic strokes were detected.

In our study, transplanted cells were distributed in the right hemisphere in the infarct area and the peri-infarct zone after IA administration. Despite the rapid elimination (within 7 days) of transplanted cells from cerebral vessels, the prolonged therapeutic effects were observed, as in the case of IV administration. The obtained data are in accordance with our previous research where was tested the influence of intra-arterially injected NPCs obtained from human MSCs in rat MCAO model (*Namestnikova et al., 2021*). In that study, short-term (2 days) presence of transplanted cells was also observed inside cerebral blood vessels which resulted in pronounced functional recovery together with the significant reduction of the infarct volume. Recently, another study of *Zhai et al. (2022)* was demonstrated that IA injection of native and pretreated with neuregulin 1β NPCs derived from human MSCs improved neurological outcome in MCAO rats. However, only administration of pretreated with neuregulin 1β NPCs led to significant reduction of infarct volume and the extent of the cerebral cortical neuron injuries and mitochondrial damage. Besides neuregulin 1β several other ways to improve the therapeutic efficacy of the intra-arterially transplanted NPCs have been made. Among them magnetic cell targeting (*Song et al., 2015*), BDNF pretreatment of injected NPCs (*Rosenblum et al., 2015*), and cell sorting for isolation of CD49d-positive NPCs (*Guzman et al., 2008*) resulted in higher cell engraftment and greater functional recovery when compared to unmodified NPCs. Nevertheless, in the present study IA transplantation of unmodified NPCs derived from iPSCs demonstrated pronounced therapeutic efficacy.

In the present study, any side effects of human iNPCs transplantation were not observed, including clinical signs of immunological rejection and teratoma formation, however, further investigations with a long-term follow-up period are needed in order to translate this technology into clinical practice.

## CONCLUSIONS

IA and IV iNPCs transplantation 24 h after stroke modeling resulted in equal degree of the improvement of function outcomes in experimental animals. However, the IA iNPCs delivery resulted in faster reduction of the stroke volume. After IA administration we observed transient trapping of iNPCs in the brain, while after IV injection no transplanted cells were found within the brain. Obtained data demonstrate that long-term presence of transplanted iNPCs in the brain is not necessary for maintaining continued functional recovery after stroke. Thus, it can be concluded that systemic transplantation of human iPSC-derived NPCs in the acute phase of ischemic stroke can be a promising therapeutic

strategy. Further studies are required in order to provide the translation of this approach into clinical practice, including assessment of mechanisms of iNPCs action within a longer observation period.

## ACKNOWLEDGEMENTS

We acknowledge the Core Center "Medical Nanobiotechnologies" of the Russian National Research Medical University for conducting the MRI diagnostics of animals.

### Funding
This work was financially supported by the Ministry of Science and Higher Education of the Russian Federation (Project No. KBK 075 0110 47 1 S7 24600 621) by "Development of New Drugs for Therapy of Neurological Diseases" The funders had no role in study design, data collection and analysis, decision to publish, or preparation of the manuscript.

### Grant Disclosures
The following grant information was disclosed by the authors:
Ministry of Science and Higher Education of the Russian Federation:  KBK 075 0110 47 1 S7 24600 621.
Development of New Drugs for Therapy of Neurological Diseases.

### Competing Interests
The authors declare there are no competing interests.

### Author Contributions

- Elvira Cherkashova performed the experiments, prepared figures and/or tables, authored or reviewed drafts of the article, and approved the final draft.
- Daria Namestnikova conceived and designed the experiments, prepared figures and/or tables, authored or reviewed drafts of the article, and approved the final draft.
- Georgiy Leonov performed the experiments, prepared figures and/or tables, and approved the final draft.
- Ilya Gubskiy performed the experiments, prepared figures and/or tables, and approved the final draft.
- Kirill Sukhinich performed the experiments, analyzed the data, prepared figures and/or tables, and approved the final draft.
- Pavel Melnikov analyzed the data, prepared figures and/or tables, and approved the final draft.
- Vladimir Chekhonin analyzed the data, authored or reviewed drafts of the article, and approved the final draft.
- Konstantin Yarygin conceived and designed the experiments, authored or reviewed drafts of the article, and approved the final draft.
- Dmitry Goldshtein conceived and designed the experiments, authored or reviewed drafts of the article, project administration, and approved the final draft.

- Diana Salikhova conceived and designed the experiments, analyzed the data, prepared figures and/or tables, authored or reviewed drafts of the article, project administration, and approved the final draft.

## Human Ethics

The following information was supplied relating to ethical approvals (i.e., approving body and any reference numbers):

The Research Centre for Medical Genetics approved the collection of human neural progenitor cells derived from human induced pluripotent cells (Protocol No. 2019-2/3 from 13 October 2020). All patients given informed written consent for using the tissue for research purposes.

## Animal Ethics

The following information was supplied relating to ethical approvals (i.e., approving body and any reference numbers):

Pirogov Russian National Research Medical University of the Ministry of Healthcare of Russian Federation provided full approval for this research

## Data Availability

The raw measurements are available in the Supplementary File.

## Supplemental Information

Supplemental information for this article can be found online at http://dx.doi.org/10.7717/peerj.16358#supplemental-information.

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
