# Peer review of "Comparative study of the efficacy of intra-arterial and intravenous transplantation of human induced pluripotent stem cells-derived neural progenitor cells in experimental stroke"

_PeerJ, doi:10.7717/peerj.16358_

## Round 0.1 · original submission · Major Revisions

Dear Authors, Major revisions needed. Thank you

Reviewer 1 ·

Basic reporting

This manuscript presented comparative study of the efficacy of intra-arterial and intravenous transplantation of human iPS-derived neural progenitor cells in experimental stroke with sufficient data and novelty. However, the problem statement/justification this study not clear. The author stated that the several clinical studies in stroke patients demonstrated general safety of the procedure and a trend toward the attenuation of the neurological deficit, and majority of the studies transplanted the NPCs intracerebrally. What is the justification to test other administration route if the current administration route is safe and can lead to improvement of neurological deficit?

Experimental design

Lacking of detailed methodologies. Need to restructure the method section.

Validity of the findings

The methodologies were not clearly written, thus, there are ambiguity in the validity of data.

Additional comments

Please check comments written on original manuscript using Track Changes tool

Annotated reviews are not available for download in order to protect the identity of reviewers who chose to remain anonymous.

Reviewer 2 ·

Basic reporting

The authors compared two important routes of cell delivery to the brain in animals with stroke. The topic of the study is very important and findings are interesting to the readership. There are however several items, which could be addressed.

• The authors were injecting arterially at an enormous speed of 100 ml/min, while it was previously reported that a speed over 1 ml/min can be damaging to the brain (PMID: 23486296). Therefore, I recommend discuss the methodology used the authors with previously published literature on the speed of injection.
• The same study (PMID: 23486296) previously pointed on the value of cell size on the safety of intra-arterial delivery, showing that for bigger MSCs overdosing is possible, while smaller GRPs can be injection at very high amounts without any risk. So, it would be critical to include the size of these NPCs, and discuss how their size corresponds to other cell types previously tested.
• In the context of the above mentioned study (PMID: 23486296) it would be worth to perform intra-arterial injections at such a high speed of 100 ml/min in intact animals and then do MRI next day to check for potential microstrokes in healthy animals, as such microstrokes could be difficult to visualize having already induced stroke in the background.
• It would be good to put into results; how many cells were labeled using each method.
• Cellular engineering using lentivector can have negative consequences, so in future it would be worth to compare engineered and non-engineered NPC in terms of behavioral and MRI outcomes.
• It would be worth in future also look into peripheral immune system, if there is any impact of NPCs on it?
• The authors used relatively insensitive mNSS test, and there was a trend toward statistical difference between IA and IV, so I would suggest to discuss that if the authors would have more tests, which would be more sensitive, then it is possible that statistical difference would be observed.
• High mortality of animals with stroke without cell transplantation is surprising, and normally it should be expected 100% survival under these conditions, and cell transplantation could only potentially improve other readouts.

Experimental design

No comment

Validity of the findings

No comments

Additional comments

No comments

Reviewer 3 ·

Basic reporting

No comment

Experimental design

No comment

Validity of the findings

Line 172: I am wondering how many GFP+ve cells were effectively transduced after 12 hours of incubation. If the data is available, the author may want to report it as well.

Figure 4: The author reported the presence of transplanted iNPCs in the rat brain decreases over time. Personally, it is preferable to report the percentage of GFP+ve cells after 2 hours and 72 hours.

Additional comments

This study claimed that systemic transplantation of induced neural progenitor cells (iNPCs) via intra-arterial and intra-venous approaches significantly improved neurological deficits in a rat model of experimental stroke, middle cerebral artery occlusion (MCAO). Within 14 days after treatment, the treatment time window after a 24-hour transplant was observed acutely and reported in detail.
Prior to transplantation, Cherkashova AE et al. derived iNPCs from human induced pluripotent stem cells (iPSC) using sufficient approaches to characterise the iPSC and iNPCs lines. Prior to being infused systemically via intra-arterial and intra-venous routes, iNPCs were labelled with GFP lentiviral vector by incubation within 12 hours, subjected to additional labelling with PKH26 Red Fluorescent labelling, and then the efficiency of the labelling was evaluated using the flowcytometry method. After 24 hours of inducing a stroke model in an animal, 7 x 105 cells/ml in PBS were transplanted using both approaches. After transplantation, the survival rate, neurological deficit, and stroke volume were thoroughly recorded and evaluated. The distribution of cells was observed 2 hours, 3 days, and 7 days after transplantation.
Later of the study, the authors concluded that both intra-arterial and intra-venous iNPC transplantation improved neurological function, but only intra-arterial transplantation decreased stroke volume. It was hypothesised that extending the observation period for this study would benefit clinical practise.
Despite the paper's simple and straightforward method, this is an important study to explore due to limited data on systematic transplantation of iNPCs. However, there is a lack of clarity in their explanation, notably in the methodology and discussion. Furthermore, I believe that this body of manuscript may be need to be improved before it is ready for publishing. Concerning the manuscript's overall technical accuracy and readability, I have only a few major/minor concerns, which are listed below:
Line 172: I am wondering how many GFP+ve cells were effectively transduced after 12 hours of incubation. If the data is available, the author may want to report it as well.
Line 242: I presume 7x105 iNSCs is referring to 7x105 iNPCs. Please clarify this information
Line 287: The author mentioned a two-way ANOVA was used to evaluate the dynamics of neurological impairment and stroke volume changes. There are missing post-doc tests utilised in this work to determine whether groups are significant, and the F statistics must also be provided.
Figure 4: The author reported the presence of transplanted iNPCs in the rat brain decreases over time. Personally, it is preferable to report the percentage of GFP+ve cells after 2 hours and 72 hours.
As such, I recommend the manuscript be submitted after addressing the comments above for a better improvement and clarity of the manuscript.

---

## Round 0.2 · Minor Revisions

Please do minor revisions thanks

Reviewer 1 ·

Basic reporting

The revised manuscript is of good quality and ready for publication despite a few minor mistakes:
1) Figure1B - did not label alpha-fetoprotein
2) Figure 1C - got two merged photos - please clarify which antibody merged with which antibody for each photo, respectively
3) There is no description about Figure 4a in the main text. Please mention the role of EBA in main text.
4) Other typing mistakes as stated in returned re-revised doc.

Experimental design

Met the criteria

Validity of the findings

Met the criteria

Additional comments

The revised manuscript is now good for publication. Only several minor mistakes to be corrected as reported in Basic Reporting section.

Annotated reviews are not available for download in order to protect the identity of reviewers who chose to remain anonymous.

Reviewer 2 ·

Basic reporting

None

Experimental design

None

Validity of the findings

None

Additional comments

None

Reviewer 3 ·

Basic reporting

No comment

Experimental design

No comment

Validity of the findings

No comment

Additional comments

The authors addressed all of my concerns with the previous version of the manuscript. The manuscript is now prepared for publication in its present form.

---

## Round 0.3 · accepted · Accept

Thank you your manuscript has been accepted.

Reviewer 1 ·

Basic reporting

The authors had answered to all my comments. I would like to suggest "Acceptance" for this manuscript.

Experimental design

No comment

Validity of the findings

No comment

Additional comments

No comment